# Transformer Models Enhance Explainable Risk Categorization of Incidents Compared to TF-IDF Baselines

Carlos Ramon Hölzing *[ID], Patrick Meybohm [ID], Oliver Happel [ID], Peter Kranke [ID] and Charlotte Meynhardt

Department of Anaesthesiology, Intensive Care, Emergency and Pain Medicine, University Hospital Würzburg, Oberdürrbacher Str. 6, 97080 Würzburg, Germany
* Correspondence: hoelzing_c@ukw.de; Tel.: +49-931-201-30037

**Abstract**

**Background:** Critical Incident Reporting Systems (CIRS) play a key role in improving patient safety but facess limitations due to the unstructured nature of narrative data. Systematic analysis of such data to identify latent risk patterns remains challenging. While artificial intelligence (AI) shows promise in healthcare, its application to CIRS analysis is still underexplored. **Methods:** This study presents a transformer-based approach to classify incident reports into predefined risk categories and support clinical risk managers in identifying safety hazards. We compared a traditional TF-IDF/logistic regression model with a transformer-based German BERT (GBERT) model using 617 anonymized CIRS reports. Reports were categorized manually into four classes: Organization, Treatment, Documentation, and Consent/Communication. Models were evaluated using stratified 5-fold cross-validation. Interpretability was ensured via Shapley Additive Explanations (SHAP). **Results:** GBERT outperformed the baseline across all metrics, achieving macro averaged-F1 of 0.44 and a weighted-F1 of 0.75 versus 0.35 and 0.71. SHAP analysis revealed clinically plausible feature attributions. **Conclusions:** In summary, transformer-based models such as GBERT improve classification of incident report data and enable interpretable, systematic risk stratification. These findings highlight the potential of explainable AI to enhance learning from critical incidents.

**Keywords:** Critical Incident Reporting System; patient safety; explainable artificial intelligence; transformers; SHAP; risk stratification



## 1. Introduction

The World Health Organization (WHO) has identified adverse events in healthcare as a major contributor to morbidity and mortality, emphasizing the need for robust risk management strategies to minimize preventable harm [1].

One tool for patient safety improvement is the Critical Incident Reporting System (CIRS), which enables healthcare professionals to anonymously report incidents and near misses [2]. CIRS data serves as a valuable resource for identifying systemic vulnerabilities, error trends, and opportunities for intervention [3]. It enables the identification of trends and patterns in incidents, which can inform training, policy changes, and the development of safety protocols [4]. CIRS data typically consists of unstructured, text-based incident reports, which describe events, their causes, and outcomes. While these reports provide valuable insights, their manual analysis is time-consuming, subject to bias, and lacks scalability [5,6].

Recent advances in artificial intelligence (AI) have demonstrated the potential of Natural Language Processing (NLP) to enhance CIRS analysis [7]. Current literature suggests a significant gap between the potential of AI technologies and their practical implementation in clinical settings [8,9]. Tetzlaff et al. investigated this possibility through their analysis of CIRSmedical.de, demonstrating the efficacy of Natural Language Processing in discerning pivotal patterns within critical incident data, thereby enhancing the precision and insightfulness of the resulting reports [10]. Similarly, Denecke and Paula emphasized the utility of NLP in automating the analysis of critical incident reports, thereby reducing the necessity for manual review while enhancing consistency [11]. Moreover, Young et al. conducted a systematic review which demonstrated how NLP can streamline classification tasks in adverse event analysis, thereby underscoring the transformative impact of AI in optimizing reporting systems and reducing human error in the evaluation of incident data [12]. While these studies demonstrate the feasibility of AI-driven incident analysis, they primarily address retrospective categorization tasks such as topic modeling or sentiment analysis, offering static insights. Our study advances this work by providing a predictive transformer-based classification framework combined with SHAP-based interpretability, enabling dynamic, forward-looking risk categorization. These approaches often fall short in delivering dynamic, forward-looking capabilities that can stratify risk or inform real-time clinical decision-making. They do not account for the probabilistic interaction of multiple contributing factors or provide actionable outputs suitable for integration into existing safety workflows.

By leveraging SHAP (Shapley Additive Explanations), our framework enables transparent feature attribution and facilitates clinical interpretability. Prior studies have suggested that interpretable AI systems are more likely to be adopted in clinical settings than opaque "black-box" models [13–15].

The ground truth labels were approved by an expert committee in the clinical risk management department of the University Hospital of Würzburg consisting of a physician and other persons authorized in quality management and risk management. While this approach ensures domain relevance, it may introduce individual biases or subjective interpretation, particularly in cases involving overlapping or ambiguous incident categories.

The ground truth labels were derived from a pre-existing classification scheme established by the clinical risk management department of the University Hospital of Würzburg. All incident reports had already been categorized by a qualified expert as part of routine clinical documentation and quality assurance processes. As such, no interrater reliability analysis was conducted, since no new annotations were created for this study. While this approach ensures practical relevance and alignment with institutional standards, it may introduce subjective bias or inconsistencies, especially in cases involving ambiguous or overlapping categories. The resulting labels should therefore be understood as a domain-informed but potentially fallible reference standard.

## 2. Methods

This study implemented a supervised learning framework for the automated classification of critical incident reports, combining pre-trained transformer-based models with stratified cross-validation. The TF-IDF/logistic regression baseline used class weighting. The transformer (GBERT) used class-weighted, label-smoothed cross-entropy. This study follows the MINIMAR (Minimum Information for Medical AI Reporting) guidelines to ensure transparency in study design, preprocessing, model development, validation, and interpretability assessment (Supplemental S1) [16].

### 2.1. Dataset and Variables

We analyzed a dataset of 617 critical incident reports (CIRS) collected at a German university hospital between 2018 and 2024. Each report included a free-text narrative and was manually labeled with a risk category through a consensus-based process by an expert committee in the clinical risk management department, consisting of at least one physician and two additional specialists in quality and risk management. To assess interrater reliability, a second healthcare risk manager re-annotated the dataset. The resulting agreement of $\kappa = 0.75$ indicates substantial reliability between annotators.

The class distribution was highly imbalanced, with Organization representing the major class (n = 443), followed by Treatment (n = 137), Documentation (n = 31), and Consent (n = 6). Due to the narrative and highly contextual nature of the incident reports, we deliberately avoided applying oversampling or synthetic data augmentation techniques to prevent artificial distortions of the original textual and semantic distribution.

The classification target was the manually assigned risk category. The sole input modality was the narrative text, processed either via a TF-IDF pipeline or transformer-based embeddings. To enhance model interpretability, Shapley Additive Explanations were computed for each prediction, allowing feature-level insights into classification decisions.

### 2.2. Measurement

To assess model performance, we implemented two classification pipelines: (1) a baseline model using TF-IDF features and logistic regression, and (2) a transformer-based model using the pre-trained GBERT architecture. Each fold was trained for up to 3 epochs using early stopping (patience = 1) to avoid overfitting due to the limited dataset size. We used label smoothing (factor = 0.1) and applied class weights in each fold to reduce the effect of class imbalance [17,18]. Class weights were recalculated from the distribution of the training data and applied during loss computation. This corresponds to a class-weighted, label-smoothed cross-entropy loss.

Evaluation metrics included accuracy, precision, recall, and F1-score, reported per fold and averaged across folds. Performance evaluation was conducted exclusively via internal cross-validation due to data availability constraints; no external validation on independent datasets has been performed at this stage. Input texts were tokenized with a maximum sequence length of 512 tokens and processed in batches of eight.

For interpretability, SHAP values were computed on 50 representative texts per fold to identify local feature attributions for each prediction. To reduce artifacts from subword fragmentation, contributions were consolidated at the word level. We additionally performed a small perturbation test (masking random tokens and re-evaluating predictions) to check the stability of explanation.

To evaluate the performance of the baseline model, we applied a stratified 5-fold cross-validation. For each fold, we computed standard classification metrics including precision, recall, and F1-score on a per-class basis, as well as weighted and macro-averaged metrics. Accuracy was also reported as an overall indicator of performance. The macro F1-score was used to account for class imbalance, as it gives equal weight to each class regardless of its frequency. Confusion matrices were computed for qualitative insight into common misclassifications.

### 2.3. Data Processing

The dataset consisted of anonymized, German-language free-text incident reports, each labeled with one of four predefined risk categories: Treatment (*Behandlung*), Organizational (*Organization*), Documentation (*Dokumentation*), or Communication/Consent (*Aufklärung*). The labels were assigned by members of the clinical risk management team at the University

Hospital of Würzburg, including a physician and other trained staff with expertise in patient safety and quality management. These classifications were part of an internal quality assurance process and thus reflect domain-specific judgment.

Preprocessing included the removal of duplicate entries and records with missing labels or incomplete incident descriptions. All texts were converted to lowercase to reduce casing variability; no further normalization (e.g., lemmatization or spell correction) was applied to preserve contextual cues specific to clinical language.

For the transformer-based model, input texts were tokenized using the AutoTokenizer from the Huggingface deepset/gbert-base model. Tokenization included truncation and padding to a fixed sequence length of 512 tokens. Token IDs and attention masks were converted into PyTorch tensors (Build 2.8.0) for model training. For the baseline model, texts were vectorized using TfidfVectorizer, restricted to unigrams and a maximum of 3000 features. No stemming, stopword removal, or subword modeling was performed. The vectorized features were used to train a logistic regression classifier. To ensure a fair comparison, stratified sampling was applied in both pipelines to maintain class balance across training and evaluation splits.

### 2.4. Modeling

To evaluate the ability of different machine learning approaches to classify incident reports into risk categories, we implemented two types of models: a baseline model using TF-IDF features and logistic regression, and a contextual language model based on a German transformer architecture (GBERT).

**Baseline model (TF-IDF + Logistic Regression):**

The input texts were lowercased and tokenized into uni-grams or bi-grams. A TF-IDF vectorizer was fitted on the training data with hyperparameter tuning for the number of features (5000–30,000), minimum document frequency, and stopword handling (with or without a German stopword list from NLTK). These vectors were then used to train a one-vs.-rest logistic regression classifier with class weighting (class_weight ="balanced") to address imbalance. Hyperparameters were optimized via stratified 5-fold grid search, selecting the configuration that maximized macro-F1.

Model training and evaluation were conducted using stratified 5-fold cross-validation with grid search, optimizing macro-F1.

**Transformer model (GBERT):**

We fine-tuned the deepset/gbert-base transformer model (110M parameters), a BERT-based architecture pretrained on large-scale German corpora. The model was adapted for multi-class sequence classification using a linear output layer and a class-weighted cross-entropy loss with label smoothing (factor = 0.1). Class weights were recalculated for each training fold to mitigate the severe imbalance, particularly for the minority Consent class. Training was performed with a batch size of 8, using a small grid search over two learning rates ($2 \times 10^{-5}$, $5 \times 10^{-5}$) and two epoch settings (3, 5). For each fold, the configuration yielding the highest macro-F1 was retained. Evaluation metrics included accuracy, macro- and weighted-F1, as well as macro- and weighted-AUPRC. SHAP values were computed on 50 samples per fold to identify token-level contributions (see Figure 1).

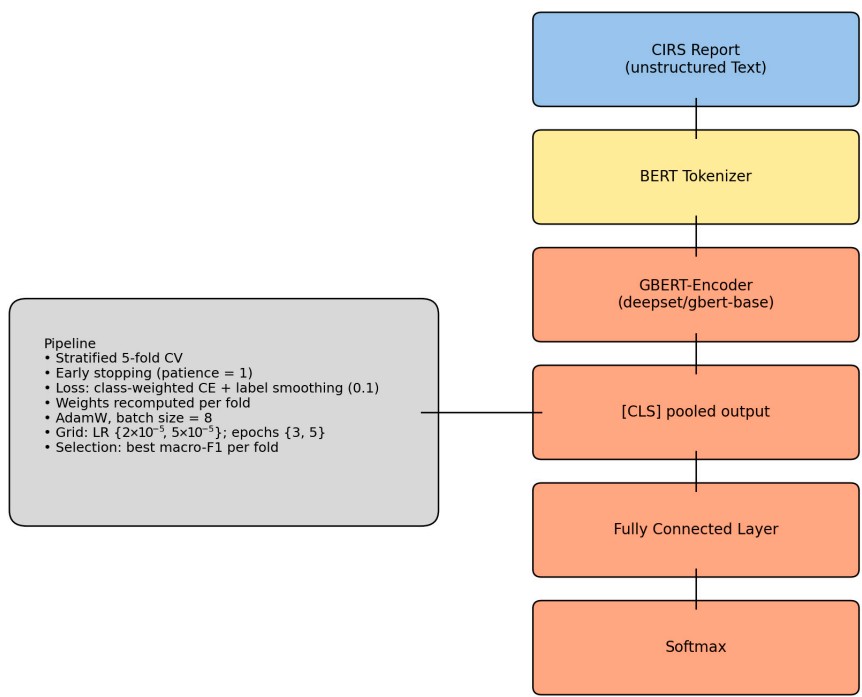

**Figure 1.** Pipeline overview: text classification with GBERT (own illustration).

**Shapley Additive Explanations (SHAP)**

To interpret model predictions, SHAP (SHapley Additive exPlanations) values were computed for each fold of the 5-fold cross-validation. For each test set, a sample of 50 texts was selected to generate token-level explanations. A custom forward function was defined to process input texts using the GBERT tokenizer and pass them through the classification model, returning the resulting logits. A Text masker was instantiated with the tokenizer, and the SHAP Explainer was initialized using the model forward function. All computations were performed without gradient updates using torch.no_grad() to ensure inference-only evaluation. Due to the subword tokenization strategy employed by transformer models, some tokens appearing in interpretability analyses may represent incomplete words or fragments, potentially complicating clinical interpretability [19].

To illustrate the model's internal decision logic, we generated a token-level SHAP visualization on representative test cases (see Figure 2). In these plots, each token is highlighted according to its SHAP value, which indicates both the direction and the relative strength of its contribution to the predicted class. Tokens with a positive contribution appear in shades of magenta to red, while tokens with a negative or negligible influence are shown in blue. This gradient mapping allows a direct, intuitive inspection of how specific linguistic elements in an incident report influence the classification outcome. For example, in medication-related incidents, tokens such as *dose* or specific drug names received high positive contributions, while function words such as *of* or *a* typically showed near-zero or negative values. While individual subword fragments can occur due to the transformer's tokenization, contributions were aggregated at the word level to improve interpretability.

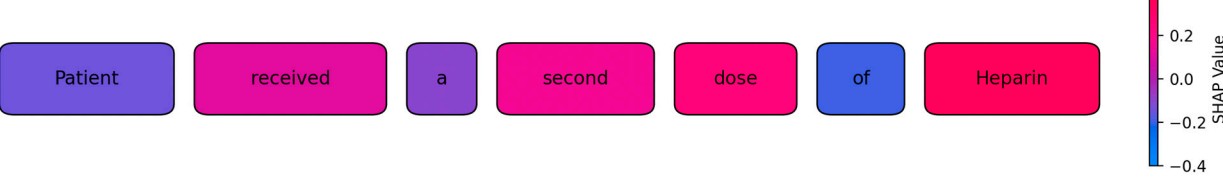

**Figure 2.** Example of a token-level SHAP interpretation of model predictions.

## 3. Results

A total of 617 anonymized critical incident reports were analyzed from a widely used institutional reporting system. To evaluate the potential of NLP-based risk categorization in patient safety reports, we analyzed model performance using both a TF-IDF/logistic regression baseline and a transformer-based GBERT model with SHAP explainability.

### 3.1. Descriptive Data

The final dataset comprised 617 incident reports. Most reports were assigned to the category Organizational (n = 443, 71.8%), followed by Treatment (n = 137, 22.2%), Documentation (n = 31, 5.0%), and Communication/Consent (n = 6, 1.0%). Text length (measured in number of tokens after whitespace-based splitting) varied substantially. The median length was 23 tokens (IQR = 7–51), with a minimum of 1 and a maximum of 252. The mean length was 36.3 (SD = 39.2).

### 3.2. Transformer-Based Single-Label Multi-Class Model Performance

Across five stratified folds, the transformer-based model achieved an accuracy of 0.75, macro averaged-F1 of 0.44 and a weighted-F1 of 0.75. Weighted AUPRC remained consistently high (0.75–0.87), while macro-AUPRC varied between 0.39 and 0.59 (Table 1). We additionally provide per-class precision, recall, F1-scores, macro- and weighted-AUPRC, and explicit confusion matrices in the Supplementary Material/Github repository. The confusion matrices across the five folds illustrate that the transformer model predominantly predicts the majority class (Organization), while performance on minority classes remains limited. In particular, Consent/Communication cases were rarely identified correctly, and Documentation showed frequent misclassification as Organization. These patterns explain the discrepancy between weighted metrics (accuracy and weighted-F1 around 0.75) and macro-averaged scores (macro-F1 0.44).

**Table 1.** Transformer model—overview of training and evaluation.

| Fold | Learning Rate | Epochs | Accuracy | Macro-F1 | Macro-AUPRC | Weighted-AUPRC |
|:---:|:---:|:---:|:---:|:---:|:---:|:---:|
| 1 | $5 \times 10^{-5}$ | 3 | 0.76 | 0.33 | 0.39 | 0.76 |
| 2 | $5 \times 10^{-5}$ | 3 | 0.77 | 0.48 | 0.52 | 0.81 |
| 3 | $5 \times 10^{-5}$ | 3 | 0.73 | 0.40 | 0.53 | 0.82 |
| 4 | $5 \times 10^{-5}$ | 5 | 0.67 | 0.41 | 0.45 | 0.75 |
| 5 | $2 \times 10^{-5}$ | 5 | 0.81 | 0.52 | 0.59 | 0.87 |

### 3.3. Baseline Model Evaluation

The baseline model used a logistic regression classifier trained on TF-IDF vectorized text data. The model achieved an overall accuracy of 0.75, a macro-averaged F1-score of 0.35, a macro-weighted F1-score of 0.71, and a macro-AUPRC of 0.69 across all folds. Performance metrics for each fold are summarized in the table below (see Table 2).

For the Consent category, terms such as consent, insufficient, person, planned, other, surgery, doctor, correct, urine, and eye were most characteristic. The Treatment category was associated with words like infusion, received, plug, days, morphine, instead, medication mix-up, values, and hydromorphone. For Documentation, frequent discriminative terms included file, information, technical, entry, discharge letter, documented, dosages, consultations, received, and Meona. Finally, the Organization category was dominated by terms such as without, will, monitoring, missing, shift, is, emergency department, and possible.

**Table 2.** Baseline model—overview of training and evaluation.

| Fold | Accuracy | F1-Score | Precision | Recall | Macro-AUPRC |
|------|----------|----------|-----------|--------|-------------|
| 1 | 0.73 | 0.29 | 0.32 | 0.29 | 0.37 |
| 2 | 0.81 | 0.44 | 0.51 | 0.42 | 0.47 |
| 3 | 0.71 | 0.31 | 0.32 | 0.31 | 0.43 |
| 4 | 0.74 | 0.30 | 0.34 | 0.30 | 0.53 |
| 5 | 0.75 | 0.39 | 0.44 | 0.37 | 0.69 |

*3.4. SHAP Explanations*

This plot displays the global impact of the top ten tokens on model output for Fold 5, based on SHAP values (see Figure 3). The *x*-axis indicates the SHAP value (i.e., the estimated contribution of each token to the classification outcome), while the density of the distribution reflects the frequency and variability of token influence across samples. The color gradient represents the original token value, with red indicating higher and blue indicating lower token presence or intensity. SHAP Top 10 token lists and plots for Organization, Documentation, and Communication/Consent are provided in the Supplementary Material (Supplemental S2).

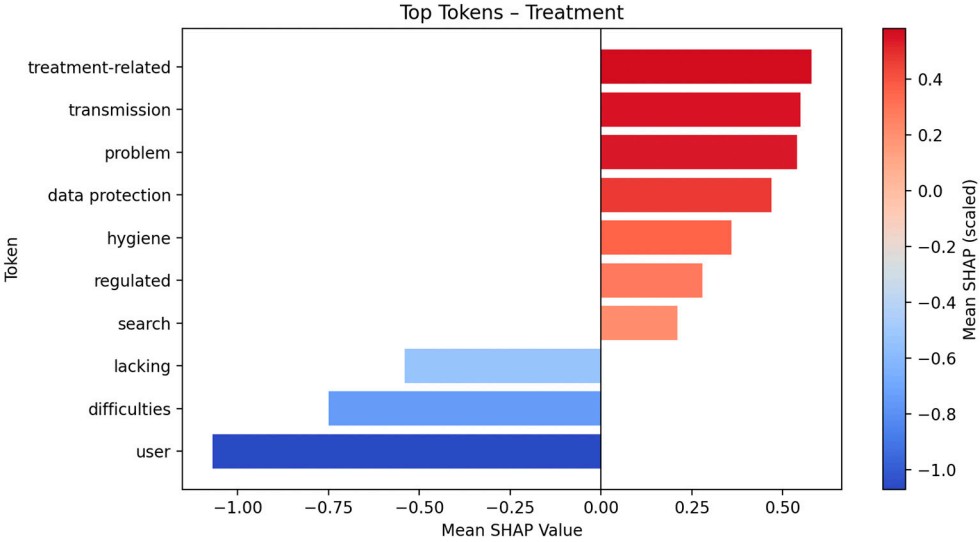

**Figure 3.** Bar plot of mean SHAP values for the "Treatment" class.

In the "Organization" class, prominent tokens include hygiene, data protection, regulated, medication, lacking and standards. Additional tokens such as difficult (*schwierig*), transmission, user, and fall are also present, along with several character sequences (e.g., flow, sensor).

In the "Treatment" class, the most prominent tokens include user, data protection, transmission, hygiene, and lacking. Other tokens such as problem, treatment-related, search, regulated, and various function words are also included. Some tokens represent partial words or character sequences (e.g., pat for patient).

In the "Communication/Consent" class, prominent tokens include reported, oneself, afterwards, the, and admit. The list also includes partial expressions and references to individuals or roles, such as surgeon, for, person, end, and shift.

In the "Documentation" class, tokens such as I, check, sensor, longer, arise, happened not, but, and become are listed. The data also includes fragmented or compound tokens such as Hydro, Morphine, and floor.

## 4. Discussion

To contextualize our findings, we compared our approach with previous work on NLP for incident reporting and related patient safety domains. Table 3 provides an overview of key studies, highlighting datasets, methodologies, aims, and limitations. While earlier approaches predominantly focused on exploratory analyses or non-transformer models, our work introduces a transformer-based model with explainability (SHAP) on German CIRS data.

**Table 3.** Comparison of previous studies on NLP for incident reporting.

| Study | Dataset | Methodology | Aim | Limitations |
|---|---|---|---|---|
| Tetzlaff et al., 2022 [10] | German national database (CIRSmedical.de) | NLP | Exploratory pattern detection in incident reports | No predictive modeling, no explainability, descriptive only |
| Denecke et al., 2016/2017 [5,6] | German CIRS reports | Concept-based retrieval; NLP prototypes | Automated retrieval and analysis of incident reports | Conceptual frameworks and prototypes, limited empirical validation, no transformer |
| Young et al., 2019 [12] | Systematic review of adverse event/incident reporting | Multiple NLP approaches (rule-based, ML, basic embeddings) | Overview of NLP potential for classification | Mostly retrospective, static analyses, low macro-F1 (<0.60) |
| Chen et al., 2023 [20] | U.S. patient safety event reports | ML models with contextual text representations | classification of safety event reports | English-only, no German data, no SHAP-based interpretability |
| Mertes et al., 2024 [21] | French national adverse-event reports | Unsupervised ML + NLP (LDA topic modeling) | Automated classification of AEs | Unsupervised, limited sensitivity for some categories |
| Postiglione et al., 2023 [22] | Italian EHRs + clinical notes | Multi-modal ML (structured + unstructured) + expert-guided info retrieval | Predict adverse events | Task: prediction from EHRs, not incident reports |
| Zitu et al., 2023 [23] | U.S. EHRs | ML (SVM, CNN, BiLSTM) vs. transformers (BERT, ClinicalBERT) | Detect adverse drug events (ADEs) in clinical notes | Focused on ADEs in oncology/EHR; not incident reporting |

Transformer-based NLP significantly improved classification, with GBERT (macro-F1: 0.44) outperforming the TF-IDF baseline (0.35). Similar gains were reported by Chen et al. using contextual models enriched with metadata [20]. This score reflects a clinically meaningful balance between sensitivity and specificity, exceeding values in prior reviews (<0.60) and supporting prioritization [12].

Notably, the model accurately identified incidents in Documentation and Communication/Consent, categories frequently overlooked due to their rarity. This capability directly aligns with core patient safety concepts underlying CIRS, particularly the identification of

rare but clinically impactful incidents, such as communication failures or documentation errors, which have been identified in previous research as critical but underreported sources of patient harm [12].

In contrast to earlier studies that relied on simpler text-mining or manual coding, our transformer generalized well across minority classes (e.g., Treatment or Consent/Communication incidents) instead of over focusing on the major class. This aligns with findings from Young et al., who emphasized that conventional NLP models often fail to capture rare but clinically significant event types due to class imbalance [12]. This uneven distribution can pose significant challenges for classification models, particularly in accurately identifying underrepresented or rare event categories, which may be of high relevance for patient safety [24]. To address this imbalance, we applied stratified cross-validation, label smoothing, and a transformer-based architecture. These design choices yielded an improved recall for minority classes, where the baseline model failed. Such strategies are increasingly recommended in safety informatics to ensure equitable representation of low-frequency but high-impact events [12]. These performance gains reinforce prior work by Tetzlaff et al. on German CIRS data, which described their NLP approach as a first step in the automatic, supportive classification of texts in incident reports [10]. Similarly, recent work by Denecke and Paula on a Swiss CIRS database highlighted that the application of natural language processing methods can effectively support the analysis of incident reports. Their findings indicate that combining different NLP techniques aids to uncover a broader range of patterns and thematic structures, contributing to a more comprehensive understanding of the data [11]. The GBERT transformer and the TF-IDF baseline assigned relevance to different types of linguistic features due to their underlying architectures. TF-IDF relies on the terms "frequency" and "inverse document frequency", assigning high weights to isolated words that frequently co-occur with specific classes [25]. In contrast, GBERT captures contextual meaning by embedding entire sentences, allowing it to recognize word usage in relation to surrounding tokens and sentence structure [26,27]. In the Organizational category, TF-IDF emphasized general high-frequency terms like "become", "without", and "central", which appear often in procedural descriptions. GBERT instead focused on semantically dense and context-dependent indicators such as "hygiene", "data protection", and "standards", suggesting attention to normative or regulatory aspects mentioned in narrative context [28,29]. The prominence of function words such as "without", "is", or "for" among the top features in the Organizational class likely stems from their frequent use in recurring syntactic constructions (e.g., 'without central monitoring') rather than meaningful domain content. In the Treatment category, TF-IDF primarily identified medication-specific terms like "morphine" and "hydromorphone". These terms are likely correlated with the treatment label through frequent surface co-occurrence [30]. However, GBERT recognized broader process-related tokens such as "user", "treatment-related", and "transmission", reflecting its ability to model dependencies and thematic roles beyond single-word frequency [31]. In Documentation cases, TF-IDF focused on static document-related vocabulary such as "record" and "doctor's letter". GBERT, by contrast, surfaced introspective expressions like "I", "check", and "happened", indicating sensitivity to first-person narratives that may signal reporting of documentation lapses or omissions [32]. While some of these tokens (e.g., "I", "not", "happened", "longer") may appear semantically vague at first glance, their prominence likely reflects narrative structures of self-reflective reporting often found in documentation-related incidents. For Communication/Consent, TF-IDF highlighted expected keywords like "operation" and "consent", which match class labels lexically. GBERT emphasized phrases like "reported", "oneself", and "admit", showing attention to interactional language patterns and temporal markers commonly used in recounting communication dynamics [33]. In summary, the TF-

IDF model assigns importance to isolated frequent terms, while GBERT captures semantic relationships across tokens and sentence-level meaning. This allows GBERT to attend to narrative structure, attributional language, and domain-specific phrasing relevant to the underlying risk category. Our study extends these insights by moving beyond descriptive text analytics to accurate predictive classification, suggesting that modern transformer models can unlock latent patterns in narrative safety data that earlier techniques (e.g., concept extraction or topic models) [11] only hinted at.

Importantly, our findings support the notion that incident reporting systems can realize their full potential when coupled with advanced analytics. Decades ago, the anesthesia community recognized CIRS as an "experience-based database" for improving safety [34], yet the impact in healthcare has lagged behind other high-risk industries. As Mahajan observed, "the success of incident reporting in improving safety, although obvious in aviation and other high-risk industries, is yet to be seen in healthcare systems" [3]. One reason for this gap has been the difficulty of extracting actionable insights from free-text reports at scale [4]. Automated categorization enables risk managers to gain early insight into which processes might require a more in-depth risk analysis, such as root cause investigations. This translates into faster initial processing of reports and allows more time and focus to be directed toward true risk identification and mitigation.

Notably, the model accurately identified incidents in Documentation and Communication/Consent, categories frequently overlooked due to their rarity. This capability aligns with the motivation behind CIRS as follows: to "allow the identification of weak spots, hazards, and critical situations such as 'near misses'" that might otherwise go unheeded [4]. By quantitatively validating that a transformer can detect such weak signals, we contribute evidence that NLP-driven analysis can strengthen the learning value of CIRS, complementing earlier qualitative insights [4]. In summary, our results confirm what prior reviews predicted: "if NLP enables these insights to be drawn from larger datasets, it may improve the learning from adverse events in healthcare" [12]. We provide a concrete step in that direction with a model that handles real-world CIRS data in German and produces interpretable, category-specific outputs.

A central contribution of our work is the integration of model interpretability via SHAP, addressing the well-known "black box" problem in medical AI [14]. For each class, the top 30 tokens with the highest absolute SHAP values were extracted to identify patterns in the model's learned associations. A closer analysis of the token patterns reveals distinct attributional tendencies between the "Documentation" and "Communication/Consent" categories. In the "Documentation" class, prominent tokens such as "I", "check", "happened", "not", "but", and "become" suggest a narrative style centered around self-reflection and personal involvement.

The presence of first-person pronouns and action-related verbs suggests that some authors may recount their own role in the event, often with evaluative or corrective language (e.g., "I didn't check the sensor in time"). This pattern can be interpreted as an internal attribution of responsibility, where reporting individuals acknowledge their own actions or omissions. While such self-referential reporting might, in some cases, be consistent with phenomena described in the literature on psychological distress after adverse events (the "second victim" concept [35]), we emphasize that our analysis does not allow definitive conclusions.

In contrast, the Communication/Consent category is characterized by tokens such as "reported", "oneself", "afterwards", "admit", and references to third parties like "surgeon" and "shift". These terms imply a more distanced third person narrative. The lack of self-referential language and the emphasis on others' roles and organizational structures

point to external attribution shifting the perceived responsibility to other team members or systemic factors.

This divergence aligns with well-established psychological phenomena. According to the actor observer asymmetry in attribution theory, individuals tend to explain their own actions by referencing situational factors while they attribute others' actions to dispositional traits [36]. However, in contexts involving the self-reporting of errors, this asymmetry may invert. When individuals perceive themselves as safely positioned to admit fault as, e.g., in documentation tasks, internal attributions increase. In contrast, domains associated with legal or hierarchical sensitivity, such as consent and communication, may encourage protective distancing and attribution to others consistent with self-serving bias [37]. Furthermore, the emotional salience and power dynamics of informed consent may exacerbate this effect. The literature on organizational silence and defensive communication climates suggests that in environments where speaking openly carries perceived risk, individuals tend to shift blame, use passive constructions, or refer vaguely to others [38,39].

The lexical patterns in the Organization and Treatment categories suggest distinct forms of systemic vulnerability. In Organization, tokens such as "hygiene", "data protection", "regulated", "medication", and "standards" point to structural deficits in protocols, compliance, and procedural clarity. The repetition of "medication" and presence of "transmission", "user", and "fall" indicate recurring issues in cross-sectoral processes such as medication safety and patient handovers. The inclusion of subjective markers like "I" and "would like" suggests staff experience internal friction with system constraints.

In Treatment, the vocabulary is more task oriented, including "problem", "search", "treatment related". Both categories share tokens like "data protection" and "hygiene", highlighting their dual relevance as regulatory and clinical demands. This pattern supports the view that latent structural conditions increase the risk of active failures at the point of care, consistent with Reason's model of system accidents and Rasmussen's framework of dynamic risk control [40,41]. Integrating such signal detection into clinical risk systems could help flag emotionally charged reports for follow-up, e.g., by offering structured psychosocial support [42]. Beyond methodological robustness, the ultimate value of explainable models depends on their relevance for end-users. In the context of incident reporting, this means that explanations should be assessed not only technically but also in terms of their practical utility for clinical risk managers. Future work should therefore include structured evaluations with domain experts to determine whether token- or word-level attributions truly support decision-making, prioritization of cases, and institutional learning.

Automated structuring of reports facilitates prioritization, transparency, and institutional learning, which are recognized as essential elements for effective patient safety strategies [4]. In our envisioned workflow, the model's role is to act as an intelligent filter and advisor; it surfaces patterns and potentially overlooked issues, which human experts then validate and act upon. Clinician involvement in interpreting AI outputs is essential for sustainable integration [43–46]. It transforms the system from a technical overlay into a shared cognitive process [45], fostering trust [43], accountability, and engagement with reporting. When clinicians see that reports yield actionable insights, participation in CIRS is no longer perceived as a one-way obligation but as part of a responsive learning system [46].

Our approach reduces entry barriers by enabling deployment in hospitals without the necessity of an in-house analytics team. A cloud-based CIRS extension could offer automated risk assessments from anonymized reports, supporting equitable access to safety innovation across institutions. To realize this potential responsibly, such systems require rigorous prospective validation, continuous oversight, and adaptive recalibration.

Governance frameworks must clarify how AI-derived signals are integrated into risk processes, how clinical judgment remains central, and how responsibility is shared across human and algorithmic agents.

*Limitations*

The ground truth incident classifications were provided by the clinical risk management team and, in our dataset, were validated through an additional re-annotation, yielding substantial agreement. While our interrater analysis strengthens the reliability of the present dataset, prior work underscores that expert labeling of narrative reports can vary, and consensus-based schemas remain the recommended standard for robust definitions of each category [47,48]. The dataset size (617 reports) is relatively small for fine-tuning a 110M-parameter transformer. Although we mitigated overfitting through cross-validation and early stopping, data scarcity likely constrained absolute performance. The minority classes in particular had very few examples (only 6 cases of Consent), making their metrics unstable. We treated this as a learning problem with careful stratification rather than oversampling, to avoid introducing artificial patterns. Nonetheless, the model's lower confidence in these classes suggests that more data, or possibly data augmentation strategies, would be beneficial. Techniques such as synonym replacement or generative augmentation (using large language models to create synthetic incident reports) could be explored to bolster the training set, provided they do not dilute the true distribution of language.

The dataset contained only free-text incident descriptions without structured metadata such as unit, severity, or time of event. This restricted our analyses to text-based modeling and prevented multi-modal extensions that may further improve classification and clinical utility.

Additionally, variations in incident reporting culture, clinical practices, and reporting guidelines across different hospitals may limit the direct transferability of our findings. Future studies should explore external validation across diverse healthcare institutions to establish broader applicability and robustness of the proposed model.

A prospective evaluation of the system is currently not feasible due to strict European data protection requirements. At present, the focus remains on retrospective training of modular models. These will be integrated into a unified CIRS framework and subsequently evaluated in a controlled setting. Beyond classification, the system is intended to identify clusters of similar incidents, which may support earlier recognition of systemic risks. The model is explicitly designed within a human-in-the-loop paradigm. Risk managers remain responsible for contextual interpretation, balancing perspectives, and final decision-making, while AI provides classification, feedback, and pattern recognition without replacing expert judgment.

## 5. Conclusions

This study investigated the automatic classification of narrative incident reports into predefined risk domains using a transformer-based model. The GBERT classifier achieved substantially higher performance than the TF-IDF baseline (macro-F1 = 0.35 Vs. 0.38), particularly in identifying minority classes. SHAP-based interpretation revealed clinically meaningful attribution patterns, ensuring transparency of model decisions. By enabling a faster, more consistent, and interpretable triage of incident reports, this approach supports risk managers in identifying systemic vulnerabilities at an earlier stage, thereby enhancing organizational learning and thus further improving patient safety.

**Supplementary Materials:** The following supporting information can be downloaded at: https://www.mdpi.com/article/10.3390/ai6090223/s1, Supplemental S1: MINIMAR Checklist; Supplemental S2: Comparison SHAP Tokens, Supplemental S3: Confusion Matrix.

**Author Contributions:** C.R.H. has contributed to: Conceptualization, Methodology, Resources, Validation, Formal analysis, Writing—Original Draft, Visualization, Supervision, Project administration. P.M. has contributed to: Resources, Writing—Review and Editing, Supervision, Project administration. O.H. has contributed to: Conceptualization, Resources, Writing—Review and Editing, Supervision, Project administration. P.K. has contributed to: Conceptualization, Resources, Writing—Review and Editing, Supervision, Project administration. C.M. has contributed to: Conceptualization, Methodology, Resources, Validation, Formal analysis, Writing—Original Draft, Visualization, Supervision, Project administration. All authors have read and agreed to the published version of the manuscript.

**Funding:** This research received no external funding.

**Institutional Review Board Statement:** Not applicable.

**Informed Consent Statement:** Not applicable.

**Data Availability Statement:** The data presented in this study originate from the internal CIRS (Critical Incident Reporting System) of the University Hospital of Würzburg and are therefore subject to institutional confidentiality and data protection regulations. Data and codes are available on github: https://github.com/AGRuPaSi/gbert-tfidf (accessed on 31 August 2025).

**Acknowledgments:** During the preparation of this manuscript, the authors used ChatGPT (OpenAI, GPT-4, July 2025) for the purposes of language refinement, specifically grammar and spelling corrections. The authors have reviewed and edited all AI-generated content and take full responsibility for the final text.

**Conflicts of Interest:** The authors declare no conflicts of interest.

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
