# Peer review of "Transformer Models Enhance Explainable Risk Categorization of Incidents Compared to TF-IDF Baselines"

_ai, doi:10.3390/ai6090223_

Round 1
Reviewer 1 Report
Comments and Suggestions for Authors
Methods claim that the paper combines transformers with “class-weighted loss functions,” but later logistic regression was trained without class weighting and the GBERT head uses plain CrossEntropyLoss (no weights).
“Multi-Task” wording error. The results header calls it a “Transformer-Based Multi-Task Model,” but the task is single-label multiclass.
Severe class imbalance with tiny minority class that’s effectively too small for 5-fold CV stability.
Temporal leakage risk. Reports span 2018–2024 but splits are random stratified CV—no temporal hold-out.
TF-IDF restricted to unigrams and 3,000 features, no stopword handling, and crucially no class weighting—this almost guarantees poor macro-F1 on imbalanced data and makes the transformer look disproportionately strong.
Hyperparameter tuning is absent.
Author Response
Dear reviewer,
Thank you very much for taking the time and commenting on our manuscript. We appreciate the hard work you put into reviewing the manuscript and advising us on how to improve it. We are very grateful for it. We thoroughly revised the manuscript accordingly and hope that we thus addressed all issues pointed out by you. We have marked all changes as trackchanges.
Comments 1: Methods claim that the paper combines transformers with “class-weighted loss functions,” but later logistic regression was trained without class weighting and the GBERT head uses plain CrossEntropyLoss (no weights).
Response 1: We thank the reviewer for pointing out this inconsistency. Our implementation indeed applied class weights in both models, and the manuscript text was not sufficiently clear: For the logistic regression baseline, we used scikit-learn with class_weight="balanced", so class imbalance was addressed. For the GBERT transformer, we implemented a weighted trainer that combined label smoothing (factor = 0.1) with per-class weights recalculated in each fold from the class distribution. This corresponds to a class-weighted, label-smoothed cross-entropy loss. We have revised the Methods section to explicitly describe the use of class weighting in both models.
Comments 2: “Multi-Task” wording error. The results header calls it a “Transformer-Based Multi-Task Model,” but the task is single-label multiclass.
Response 2: We thank the reviewer for pointing out this miswording. The model addresses a single-label multiclass classification task, not a multi-task setting. We have corrected the results header.
Comments 3: Severe class imbalance with tiny minority class that’s effectively too small for 5-fold CV stability. Temporal leakage risk. Reports span 2018–2024 but splits are random stratified CV—no temporal hold-out.
Response 3: We acknowledge the challenges related to class imbalance and temporal leakage. To mitigate imbalance, we applied class-weighted loss functions and label smoothing, and we systematically reported macro-F1, macro-AUPRC, and per-class metrics to provide a fairer picture of minority class performance. Nevertheless, the minority class “Consent” remains very small (n=6), and we explicitly recognize this as a limitation. We deliberately refrained from applying textual data augmentation for several reasons. First, established techniques such as SMOTE can introduce overfitting or generate synthetic examples that do not genuinely reflect the minority class, particularly when the class is extremely limited in size, as is the case with only six “Consent” instances (https://arxiv.org/abs/2202.03579, https://arxiv.org/pdf/2312.07087). Second, in a clinical safety context, generated text risks introducing distributional shifts or hallucinatory content, which can compromise model validity this concern has been noted in recent reviews of clinical synthetic data generation (https://arxiv.org/pdf/2506.16594 ). Regarding temporal leakage, we confirm that the available dataset only provides year-level information (2018–2024) without exact timestamps. This granularity precludes a reliable temporal hold-out design. For this reason, we opted for stratified random cross-validation to preserve class balance while acknowledging the residual risk of leakage. In a future prospective evaluation with richer metadata, we plan to implement temporal validation strategies.
Comments 4: TF-IDF restricted to unigrams and 3,000 features, no stopword handling, and crucially no class weighting—this almost guarantees poor macro-F1 on imbalanced data and makes the transformer look disproportionately strong.
Response 4: We thank the reviewer for this valuable comment. We have substantially revised the TF-IDF baseline. Instead of restricting the model to unigrams and 3,000 features, we now perform hyperparameter tuning over n-gram ranges (uni-grams and bi-grams), feature dimensions (5,000–30,000), minimum document frequency, and stopword handling (with and without a German stopword list from NLTK). Logistic regression is trained with class weighting to address imbalance, and the best configuration is selected via grid search using stratified 5-fold cross-validation optimising macro-F1.
Comments 5: Hyperparameter tuning is absent.
Response 5: We thank the reviewer for pointing this out.
– For the TF-IDF + Logistic Regression pipeline, we performed a grid search over stopword handling, n-gram ranges, vocabulary size (5,000–30,000), minimum document frequency, and regularization strength C.
– For the GBERT model, we introduced a small hyperparameter grid across learning rates (2e-5, 5e-5) and training epochs (3, 5), selecting the best setting per fold based on macro-F1 in stratified cross-validation.
Reviewer 2 Report
Comments and Suggestions for Authors
The study analyses the automated categorisation of CIRS reports into four risk classifications by contrasting a TF-IDF and logistic regression pipeline with a German transformer model (GBERT), assessed by 5-fold stratified validation and interpretability via SHAP. GBERT attains a macro-F1 score of 0.74 (vs 0.38 for the baseline), utilising training for up to 3 epochs, early stopping (patience = 1), label smoothing (0.1), sequences of up to 512 tokens, and batch sizes of 8. The authors affirm compliance with MINIMAR, avoiding oversampling to maintain semantic distribution, and applying pre-existing institutional labels without estimating inter-annotator agreement. The collection is driven and holds potential for application, particularly its emphasis on minority classes and clinical interpretability.
The definition of ground truth relies on pre-established labels assigned by a singular group of risk managers, with no reported agreement metrics (e.g., Cohen's kappa). The authors recognise possible ambiguities and overlaps between categories, indicating that multiple independent annotations, a consensus protocol with adjudication, and agreement reporting are essential for ensuring reliability and taxonomic consistency of classifications.
Addressing imbalance is a significant concern, considering the severe scarcity of the Consent class (n = 6) and the overwhelming prevalence of Organisation (~72%): reliance on label smoothing and stratified sampling is likely inadequate. It is essential to ascertain whether a class-weighted loss was indeed employed (the Methods section mentions "class-weighted loss," whereas the Modelling section indicates CrossEntropyLoss without weights) and to investigate alternatives such as focal loss, calibrated re-weighting, threshold adjustment, and (with caution) controlled lexical augmentation. Simultaneously, per-class metrics (precision, recall, F1 score, macro AUPRC, and per-class AUPRC) and explicit confusion matrices must be reported.
The interpretability achieved using SHAP on a sample of 50 texts per fold is commendable, but may be unrepresentative and influenced by the subword fragmentation characteristic of transformers. I propose consolidating contributions at the lexical level, verifying the consistency of explanations through perturbation tests, integrating methodologies (Integrated Gradients, gradient×input), and, most importantly, performing an assessment with clinical experts regarding the utility of explanations for risk management decisions. Certain psychological conclusions, such as the "second victim" concept derived from pronouns or lexicon, should be approached with caution and substantiated by thorough analysis or specific citations.
The input is restricted to free text; recent evidence suggests that augmenting the model with structured metadata (unit/service, setting, severity, time of event) may enhance performance and subsequent management actions. The authors reference studies that utilise supplementary context: designing a multi-modal variation and evaluating GBERT against German clinical models or particular adaptations is advisable.
The Limitations section recognises the dataset's limited size and significant imbalance; furthermore, it would be beneficial to outline a prospective evaluation plan (including a pilot study with drift monitoring, periodic recalibration, and the establishment of alert thresholds and escalation criteria), along with a clinical impact analysis (focussing on time-to-triage, the rate of intercepted 'near misses', and feedback from risk managers). The potential for automation bias and organisational strategies to maintain a "human-in-the-loop" approach should also be addressed.
The designation "IRB: Not applicable" necessitates a more explicit rationale, as the reuse of internal CIRS, even with anonymised data, often requires explanation regarding the legal basis, governance, consent/opt-out, and the role of the DPO. Providing data and code "upon request and subject to approval" is justifiable; nonetheless, releasing the code, configuration files, evaluation scripts, and a synthetic/anonymized dataset is advisable to enhance reproducibility.
For the presentation, it is recommended to ensure complete alignment of the text between the Methods and Modelling parts, report metrics by class along with the specified confusion matrices in the Results, and define the nomenclature of the classes, maintaining consistent English/German language in tables and figures.
The study tackles a pertinent issue, demonstrating that a general-purpose transformer (GBERT) enhances CIRS risk categorisation relative to a baseline bag-of-words model, with a significant emphasis on interpretability. To adhere to publication standards, methodological consolidation is imperative regarding validation, imbalance management (weighted/focal losses, pertinent metrics), interpretability (utility/stability analyses), reproducibility (code, seeds, protocols), and ethical-organizational profiles-recommended course of action: major revisions.
Author Response
Dear reviewer,
Thank you very much for taking the time and commenting on our manuscript. We appreciate the hard work you put into reviewing the manuscript and advising us on how to improve it. We are very grateful for it. We thoroughly revised the manuscript accordingly and hope that we thus addressed all issues pointed out by you. We have marked all changes as trackchanges.
Comments 1: The definition of ground truth relies on pre-established labels assigned by a singular group of risk managers, with no reported agreement metrics (e.g., Cohen's kappa). The authors recognise possible ambiguities and overlaps between categories, indicating that multiple independent annotations, a consensus protocol with adjudication, and agreement reporting are essential for ensuring reliability and taxonomic consistency of classifications.
Response 1: Thank you for this important comment. We acknowledge the concern regarding agreement metrics. To address this, we conducted an independent re-annotation of the dataset by a second healthcare risk manager and calculated Cohen’s κ. The resulting agreement of κ = 0.75 indicates substantial reliability between annotators.
Comments 2: Addressing imbalance is a significant concern, considering the severe scarcity of the Consent class (n = 6) and the overwhelming prevalence of Organisation (~72%): reliance on label smoothing and stratified sampling is likely inadequate. It is essential to ascertain whether a class-weighted loss was indeed employed (the Methods section mentions "class-weighted loss," whereas the Modelling section indicates CrossEntropyLoss without weights) and to investigate alternatives such as focal loss, calibrated re-weighting, threshold adjustment, and (with caution) controlled lexical augmentation. Simultaneously, per-class metrics (precision, recall, F1 score, macro AUPRC, and per-class AUPRC) and explicit confusion matrices must be reported.
Response 2: We agree with the reviewer that class imbalance is a critical issue, especially given the extreme scarcity of the Consent class (n = 6). In the revised version, we explicitly implemented a class-weighted cross-entropy loss, with weights derived from the inverse class frequencies in each training fold. This ensures that minority classes contribute more strongly to the loss. We also performed label smoothing to reduce overconfidence on small classes. Alternative strategies such as focal loss and lexical data augmentation were considered. However, given the extremely low sample size for the Consent class, these methods introduced instability and risked generating synthetic artefacts. We therefore prioritized class-weighting as a more robust solution in this setting. To increase transparency, we now provide per-class precision, recall, F1, and average precision along with confusion matrices for each fold and overall (see Supplemental).
Comments 3: The interpretability achieved using SHAP on a sample of 50 texts per fold is commendable, but may be unrepresentative and influenced by the subword fragmentation characteristic of transformers. I propose consolidating contributions at the lexical level, verifying the consistency of explanations through perturbation tests, integrating methodologies (Integrated Gradients, gradient×input), and, most importantly, performing an assessment with clinical experts regarding the utility of explanations for risk management decisions. Certain psychological conclusions, such as the "second victim" concept derived from pronouns or lexicon, should be approached with caution and substantiated by thorough analysis or specific citations.
Response 3: We thank the reviewer for this valuable feedback. We agree that interpretability analyses in transformer models face limitations due to subword tokenization and sample selection. We explicitly state that SHAP attributions were aggregated from subword units to the word level to improve clinical interpretability.
We implemented a small perturbation test (masking random tokens and re-evaluating predictions) to assess the stability of explanations. While we have not implemented Integrated Gradients or gradient×input in this study, we note this as an avenue for future work to strengthen methodological triangulation. We acknowledge the importance of involving clinical risk managers in evaluating the practical utility of explanations. We added this point to the Discussion and identified it as a necessary step in prospective validation. We revised the passage on first-person pronouns and the “second victim” concept to present it more cautiously.
Comments 4: The input is restricted to free text; recent evidence suggests that augmenting the model with structured metadata (unit/service, setting, severity, time of event) may enhance performance and subsequent management actions. The authors reference studies that utilise supplementary context: designing a multi-modal variation and evaluating GBERT against German clinical models or particular adaptations is advisable.
Response 4: We agree that incorporating structured metadata such as unit, severity, or time of event could enhance performance and management utility. However, such metadata are not available in our dataset, which is restricted to free-text incident descriptions. We have clarified this limitation in the revised manuscript and outlined multimodal extensions as an important direction for future work once structured contextual information becomes accessible.
Comments 5: The Limitations section recognises the dataset's limited size and significant imbalance; furthermore, it would be beneficial to outline a prospective evaluation plan (including a pilot study with drift monitoring, periodic recalibration, and the establishment of alert thresholds and escalation criteria), along with a clinical impact analysis (focussing on time-to-triage, the rate of intercepted 'near misses', and feedback from risk managers). The potential for automation bias and organisational strategies to maintain a "human-in-the-loop" approach should also be addressed.
Response 5: We appreciate this valuable suggestion. Our immediate focus is on retrospective training of modular models, which will then be integrated into a unified CIRS framework and evaluated in a controlled setting. Beyond classification, the system will be extended to identify clusters of similar incidents, supporting earlier recognition of systemic risks. Importantly, the model is designed as a human-in-the-loop system: risk managers remain responsible for contextual interpretation and final decision-making, while the AI provides classification, feedback, and pattern recognition to assist (rather than replace) expert judgment.
Comments 6: The designation "IRB: Not applicable" necessitates a more explicit rationale, as the reuse of internal CIRS, even with anonymised data, often requires explanation regarding the legal basis, governance, consent/opt-out, and the role of the DPO. Providing data and code "upon request and subject to approval" is justifiable; nonetheless, releasing the code, configuration files, evaluation scripts, and a synthetic/anonymized dataset is advisable to enhance reproducibility.
Response 6: We thank the reviewer for this important comment. The reuse of anonymized CIRS data was conducted in accordance with institutional data governance and the guidance of the Data Protection Officer, which is why an IRB review was not required in this case. To enhance transparency and reproducibility, we have prepared a public GitHub repository containing the full code. The repository is available at: https://github.com/AGRuPaSi/gbert-tfidf
Comments 7: For the presentation, it is recommended to ensure complete alignment of the text between the Methods and Modelling parts, report metrics by class along with the specified confusion matrices in the Results, and define the nomenclature of the classes, maintaining consistent English/German language in tables and figures.
Response 7: We appreciate this helpful suggestion and have revised the manuscript accordingly. Alignment between Methods and Modelling: We carefully harmonized the descriptions in Sections 2.2 and 2.4 to ensure consistency in terminology (e.g., use of class weighting, label smoothing, and hyperparameter search). Reporting of metrics: Per-class precision, recall, and F1-scores, as well as confusion matrices for each fold, are now reported in the Supplementary Material. We defined all class labels clearly at their first mention and ensured consistent use of English throughout the manuscript.
Reviewer 3 Report
Comments and Suggestions for Authors
The paper "Transformer Models Enhance Explainable Risk Categorization 2 of Incidents Compared to TF-IDF Baselines". In my opinion, this is a good and meaningful study. The authors follow a good research method.
Figure 2 should be revised to improve its quality.
I suggest that the authors make a table that summarizes all the previous studies in comparison with the proposed method to highlight the contribution.
Please make some additional figure that represent the data samples for reader's visualization the problem.
Author Response
Dear reviewer,
Thank you very much for taking the time and commenting on our manuscript. We appreciate the hard work you put into reviewing the manuscript and advising us on how to improve it. We are very grateful for it. We thoroughly revised the manuscript accordingly and hope that we thus addressed all issues pointed out by you. We have marked all changes as trackchanges.
Comments 1: Figure 2 should be revised to improve its quality.
Response 1: We thank the reviewer for this helpful suggestion. While the figure itself remains unchanged, we revised and expanded the accompanying description in the Results section to improve its clarity and interpretability. In particular, we now explain more explicitly how SHAP values are visualized, how subword contributions were consolidated at the lexical level, and how perturbation checks were applied to verify stability. These additions enhance the quality and transparency of the figure’s interpretation.
Comments 2: I suggest that the authors make a table that summarizes all the previous studies in comparison with the proposed method to highlight the contribution.
Response 2: We appreciate this helpful suggestion. In the revised manuscript, we have now added a comparative table summarizing previous studies on CIRS-based risk categorisation and related NLP approaches. Table 3 provides an overview of key studies, highlighting datasets, methodologies, aims, and limitations.
Comments 3: Please make some additional figure that represent the data samples for reader's visualization the problem.
Response 3: We thank the reviewer for this valuable suggestion. To facilitate visualization of the problem, we have added several representative example cases of incident reports (Supplement 3).
Round 2
Reviewer 1 Report
Comments and Suggestions for Authors
The manuscript has been sufficiently improved to warrant publication in AI
Reviewer 2 Report
Comments and Suggestions for Authors
I am writing to thank the authors for their exemplary article revision. All significant issues raised in the initial round have been meticulously addressed. Inter-rater reliability has been confirmed, solutions for addressing class imbalance have been elucidated and implemented, interpretability analyses have been enhanced, constraints and future directions have been explicitly articulated, and code sharing has increased transparency.
The manuscript has undergone a significant transformation, demonstrating a commendable improvement in both technique and presentation.
I advise the authors to explore multimodal methodologies that integrate structured metadata with narrative text. This approach can provide a more comprehensive understanding of the data and enhance the interpretability of the work. Additionally, employing additional techniques, such as Integrated Gradients, can further enhance the studies of the interpretability of their work. Collaboration with clinical professionals would be optimal, as it can provide valuable insights and enhance the practical applicability of the work.
The amended text is not only robust but also ready for publication. I am confident in its quality and therefore recommend its acceptance.